# From spikes to intercellular waves: Tuning intercellular calcium signaling dynamics modulates organ size control

**Dharsan K. Soundarrajan**[1º], **Francisco J. Huizar**[1,2º], **Ramezan Paravitorghabeh**[1º], **Trent Robinett**[1], **Jeremiah J. Zartman**[1,2]*

**1** Department of Chemical and Biomolecular Engineering, University of Notre Dame, South Bend, Indiana, United States of America, **2** Bioengineering Graduate Program, University of Notre Dame, South Bend, Indiana, United States of America

º These authors contributed equally to this work.

* jzartman@nd.edu

**Data Availability Statement:** All the data and simulation codes are available on our lab GitHub repository. On the linked webpage below are

## Abstract

Information flow within and between cells depends significantly on calcium ($Ca^{2+}$) signaling dynamics. However, the biophysical mechanisms that govern emergent patterns of $Ca^{2+}$ signaling dynamics at the organ level remain elusive. Recent experimental studies in developing *Drosophila* wing imaginal discs demonstrate the emergence of four distinct patterns of $Ca^{2+}$ activity: $Ca^{2+}$ spikes, intercellular $Ca^{2+}$ transients, tissue-level $Ca^{2+}$ waves, and a global "fluttering" state. Here, we used a combination of computational modeling and experimental approaches to identify two different populations of cells within tissues that are connected by gap junction proteins. We term these two subpopulations "initiator cells," defined by elevated levels of Phospholipase C (PLC) activity, and "standby cells," which exhibit baseline activity. We found that the type and strength of hormonal stimulation and extent of gap junctional communication jointly determine the predominate class of $Ca^{2+}$ signaling activity. Further, single-cell $Ca^{2+}$ spikes are stimulated by insulin, while intercellular $Ca^{2+}$ waves depend on $Gαq$ activity. Our computational model successfully reproduces how the dynamics of $Ca^{2+}$ transients varies during organ growth. Phenotypic analysis of perturbations to $Gαq$ and insulin signaling support an integrated model of cytoplasmic $Ca^{2+}$ as a dynamic reporter of overall tissue growth. Further, we show that perturbations to $Ca^{2+}$ signaling tune the final size of organs. This work provides a platform to further study how organ size regulation emerges from the crosstalk between biochemical growth signals and heterogeneous cell signaling states.

## Author summary

Calcium ($Ca^{2+}$) is a universal second messenger that regulates a myriad of cellular processes such as cell division, cell proliferation and apoptosis. Multiple patterns of $Ca^{2+}$ signaling including single-cell spikes, multicellular $Ca^{2+}$ transients, large-scale $Ca^{2+}$ waves, and global "fluttering" have been observed in epithelial systems during organ

instructions on how to run a sample demonstration of the code on Google Colaboratory and how to run the code on a personal computer. Simulations corresponding to the main text figure conclusions were repeated five separate times with five different random number generator seeds (S13 Fig). This was done to ensure reproducibility of the drawn conclusions, and the corresponding outputs are documented in the GitHub repository and the linked GitHub webpage: https://multicellularsystemslab.github.io/MSELab_Calcium_Cartography_2021/.

**Funding:** The work in this paper was supported by NIH Grant R35GM124935 (JJZ) and NSF Award CBET-1553826 (JJZ). The funders had no role in study design, data collection and analysis, decision to publish, or preparation of the manuscript.

**Competing interests:** The authors have declared that no competing interests exist.

development. Key molecular players and biophysical mechanisms involved in formation of these patterns during organ development are not well understood. In this work, we developed a generalized multicellular model of $Ca^{2+}$ that captures all the key categories of $Ca^{2+}$ activity as a function of key hormonal signals. Integration of model predictions and experiments reveals two subclasses of cell populations and demonstrates that $Ca^{2+}$ signaling activity at the organ scale is defined by a general decrease in gap junction communication as an organ grows. Our experiments also reveal that a "goldilocks zone" of optimal $Ca^{2+}$ activity is required to achieve optimal growth at the organ level.

## Introduction

Mechanisms of intercellular communication are critical during epithelial morphogenesis when cells communicate and coordinate their activities to generate functioning organs [1,2]. One modality of intercellular communication occurs through gap junctions (GJ), intercellular channels that permit direct cell-cell transfer of ions and other small molecules [3]. Calcium ions ($Ca^{2+}$) act as second messengers that regulate a myriad of cellular processes such as proliferation, differentiation, transcription, metabolism, cellular motility, fertilization, and neuronal communication [4–13]. $Ca^{2+}$ signaling also regulates developmental processes at the multicellular level. For instance, $Ca^{2+}$ signaling has been shown to regulate scale development in butterfly wings [14]. It also mediates autophagic and apoptotic processes required for hearing acquisition in the developing cochlea [15–17]. However, a systems-level description of $Ca^{2+}$ signaling during organ development is lacking.

A major challenge in reverse engineering $Ca^{2+}$ signaling during organ development is the lack of an in vivo model system to identify how cells interpret and integrate information across the broad range of input molecules that dynamically vary concentrations of cytosolic $Ca^{2+}$ ions. In particular, it remains unclear how single-cell $Ca^{2+}$ dynamics are coordinated to regulate tissue-level $Ca^{2+}$ patterns. To overcome these challenges, we developed a computational model based on a realistic geometry of epithelial cells to model $Ca^{2+}$ signaling in the *Drosophila* wing imaginal disc. The *Drosophila* wing imaginal disc is an experimentally amenable system for investigating systems-level regulation of cell signaling (Fig 1A) [18–20]. Overall, *Drosophila* wing imaginal discs are a premier system to gain insights into several organ-intrinsic and organ-extrinsic mechanisms that control organ growth [21–26].

Previous experimental investigations revealed four classes of $Ca^{2+}$ signaling activity in the developing wing disc: single-cell $Ca^{2+}$ spikes, multicellular transients, intercellular waves, and global fluttering (Fig 1B and 1C). We have shown recently that the patterns depend on the strength of agonist stimulation [19]. We and others have previously reported that $Ca^{2+}$ patterns observed in the wing disc are dependent on phospholipase C (PLC) and the inositol trisphosphate receptor ($IP_3R$) pathway mediated by gap junctional communication [18–20]. In non-excitable cells, stimulation of receptors in the cell surface results in activation of PLCs to generate $IP_3$, which binds to and activates $IP_3R$ (Fig 1D) [6]. Upon binding, $IP_3Rs$ channel $Ca^{2+}$ from the endoplasmic reticulum (ER) to the cytosolic space [6,27,28]. However, the specific receptors involved in stimulation of PLCs in the *Drosophila* wing imaginal disc remain to be more fully defined. The key physical/chemical parameters and their interactions that define multicellular $Ca^{2+}$ dynamics in response to agonist stimulation is not fully characterized. How these different spatiotemporal modes of signaling encode information from upstream signals that impact downstream cellular processes during organ development is poorly understood. We have previously shown that inhibition of $Ca^{2+}$ regulators of $IP_3$, including PLC21C, Gαq

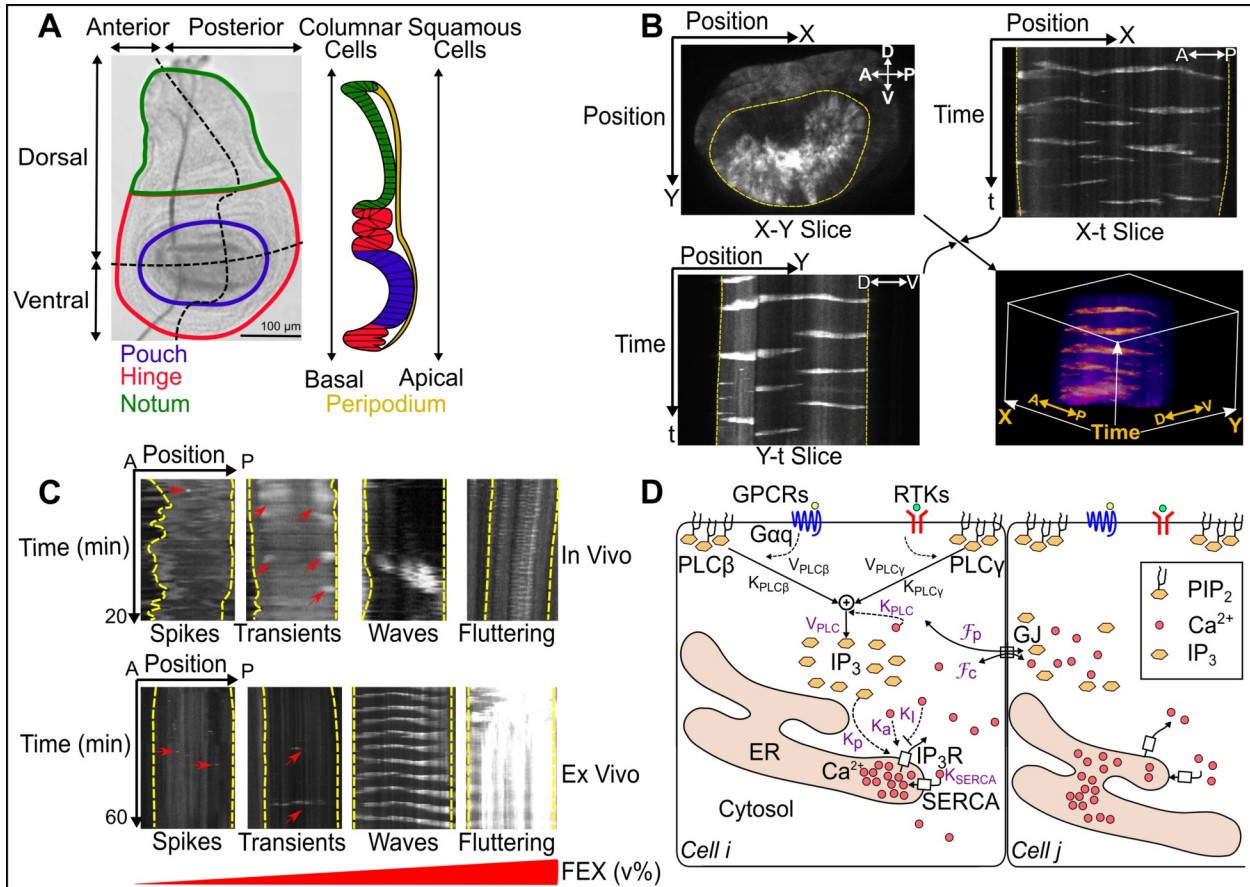

**Fig 1. Multicellular Ca²⁺ signaling in a developing organ.** **(A)** (*Left panel*) Image of third instar *Drosophila* wing imaginal disc. The larval wing disc includes main four regions: pouch (blue), hinge (red), notum (green) and peripodium (yellow). The pouch cells are the region of interest for this study. (*Right panel*) A schematic of the side view of the wing disc showing the peripodial membrane composed of squamous epithelial cells. **(B)** Kymographs illustrate two-dimensional slices of three-dimensional (X, Y and t planes respectively) spatiotemporal signaling. Ca²⁺ signaling activity is related to the fluorescence intensity of the Ca²⁺ sensor, GCaMP6f. A view of the X-Y plane (top left), the X-t plane (top right), and Y-t plane (bottom left) are combined to illustrate a 3D view of the signaling activity (bottom right). Yellow dashed lines indicate the pouch region. X coordinates roughly correspond to the A/P direction of the wing disc pouch whereas the Y coordinate principally describes the D/V axis of the pouch. **(C)** Four classes of Ca²⁺ signaling patterns are observed both in vivo and ex vivo: single-cell spikes, intercellular transient, intercellular waves and fluttering [19]. In ex vivo cultures, the occurrence of these patterns depend on the concentration of fly extract added to the culture media. Red arrows highlight subtle cellular Ca²⁺ activity. **(D)** Major components of Ca²⁺ toolkit: G protein–coupled receptors (GPCRs), receptor tyrosine kinase (RTKs), gap junctions (GJ), Inositol trisphosphate (IP₃), diacylglycerol (DAG), Phospholipase C (PLC), Phosphatidylinositol 4,5-bisphosphate (PIP₂), sarco/endoplasmic reticulum Ca²⁺-ATPase (SERCA) and IP₃ receptors (IP₃R). Parameters for our computational model are denoted in purple. $K_{PLC}$ and $V_{PLC}$ values are lumped activity parameters that are describe the stimulus-dependent activity of specific PLC isoforms, specifically PLCγ downstream of the insulin receptor and PLCβ downstream of GPCR signaling.

and the gap junction protein Innexin 2 (Inx2) in the wing disc results in reduction in size of adult wing blade [19]. Whether changes to multiscale Ca²⁺ signaling patterns in wing disc alters overall adult blade wing size remains unknown.

Overall, the computational models of calcium signaling in developing epithelial systems have received sparse attention to date. Here, we report the necessary conditions to generate the full spectrum of experimentally observed spatiotemporal patterns by employing a computational modeling approach. To do so, we built a geometrically accurate 2D-model of a wing disc based on experimental data. We discovered that in silico replication of wing disc Ca²⁺ patterns requires two distinct classes of cells, which we term as "initiator cells" and "standby cells." Here, we show how the standby cells organize themselves with respect to a Hopf

bifurcation threshold of the model's $V_{PLC}$ parameter, and how the range of standby cell $V_{PLC}$ values determine the final patterns of $Ca^{2+}$ signaling. Next, computational simulations and experiments demonstrate that gap junction communication alters $Ca^{2+}$ signaling response resulting in more $Ca^{2+}$ spikes in the absence of external stimuli. Finally, we provide computational and experimental evidence for the role of $Ca^{2+}$ signaling in imaginal disc morphogenesis. Our findings suggest a "goldilocks zone" of integrated $Ca^{2+}$ signaling where lower levels of $Ca^{2+}$ is correlated with reduced organ growth and higher levels of $Ca^{2+}$ is also correlated with reduced growth dependent upon the stimulus triggering the $Ca^{2+}$ signal. Overall, we identify crucial crosstalk between biochemical growth signals, such as insulin and Gαq, and heterogeneous cell signaling states during the growth of an organ.

## Methods

### Computational model

Several mathematical models have been proposed to describe intra- and intercellular $Ca^{2+}$ wave propagation [29]. Here we extended a previously formulated model reported by Politi and colleagues that described single-cell $Ca^{2+}$ oscillations observed in Chinese hamster ovarian (CHO) cells [30]. This model serves as a baseline model in cells for $IP_3R$-mediated $Ca^{2+}$ signaling in the *Drosophila* wing disc. This model accounts for the formation and degradation of $IP_3$, $Ca^{2+}$ flux across the endoplasmic reticulum (ER) through $IP_3R$, and sarco/endoplasmic reticulum $Ca^{2+}$-ATPase (SERCA), $IP_3R$, and ER $Ca^{2+}$ dynamics. The model consists of four state variables: cytosolic $IP_3$ ($p$), cytosolic $Ca^{2+}$ ($c$), the ER $Ca^{2+}$ concentration ($s$), and the fraction of $IP_3$ receptors that have not been inactivated by $Ca^{2+}$ ($r$).

### IP_3 dynamics

$IP_3$ is generated in the cytosol by phospholipases (PLC) [31]. The *Drosophila* genome consists of three PLC genes. They include *PLC21C* and *norpA*, which are related to the PLCβ1–4 subfamily of *Homo sapien* homologs, and a single PLCγ (*sl*) [32]. While different classes of PLC can hydrolyze $PI(4,5)P_2$ to generate $IP_3$ and DAG, they are activated by different receptors on the cell surface. For instance, PLCβ homolog PLC21C is activated by the heterotrimeric G-protein αq subunit in response to G-protein receptor signaling [33]. On the other hand, PLCγ is recruited via its SH2 domain to activated receptor tyrosine kinase, such as the insulin receptor, at the plasma membrane [32]. In our model, we describe all the combined production of $IP_3$ as dependent on the total PLC activity of the cell:

$$v_{PLC_i} = V_{PLC} \frac{c_i^2}{K_{PLC}^2 + c_i^2}, \tag{1}$$

where $V_{PLC}$ describes the maximal production rate of $IP_3$, and $K_{PLC}$ describes the sensitivity of PLC to $Ca^{2+}$. The parameter $V_{PLC}$ depends on agonist concentration, we assume that $V_{PLC}$ describes a summed activity of PLCs activated by upstream receptors.

Our model also considers degradation of $IP_3$ by other factors such as $IP_3$ kinases, which converts $IP_3$ to $IP_4$. We generalize the degradation of $IP_3$ using first order kinetics. Collectively, the equation describing the dynamics of $IP_3$ in a cell is:

$$\frac{dp_i}{dt} = J_{p_i} + v_{PLC_i} - k_{5,P} p_i, \tag{2}$$

where $J_{p_i}$ is the flux of cell $i$ $IP_3$ through gap junction communication and $k_{5,P}$ is the $IP_3$ dephosphorylation rate constant. We assume that $IP_3$ diffuses from one cell to the adjacent

cells through gap junctional coupling. We model the flux through gap junctions (GJs) using the following equation:

$$J_{p_i} \approx F_p \left[ \sum_{j \in N_i} p_j l_{ij} - p_j \left( \sum_{j \in N_j} l_{ij} \right) \right],$$

(3)

where $F_p$ refers to the IP$_3$ permeability of the gap junctions, $p_j$ refers to the IP$_3$ concentration in neighboring cell $j$ and $l_{ij}$ refers to the length of cell boundary shared by cells $i$ and $j$ respectively. We assume that the intracellular diffusion of IP$_3$ is fast relative to the diffusion of IP$_3$ between cells through GJs. Consequently, we have neglected terms that describe intracellular diffusion.

## Ca$^{2+}$ dynamics

Ca$^{2+}$ is released through IP$_3$R from the ER. Similarly, cytosolic Ca$^{2+}$ is pumped into the ER using SERCA pumps. In many cell types, Ca$^{2+}$ is also pumped out from the cytosol to the extracellular space through the plasma membrane. In our model, we ignore the flux of Ca$^{2+}$ through the cell's plasma membrane, and we only consider the transport of Ca$^{2+}$ from the ER to cytosol. This assumption is based on our previous experimental studies showing that the observed Ca$^{2+}$ dynamics in the wing disc are due to the IP$_3$ mediated Ca$^{2+}$ release through IP$_3$R from ER [18,19]. To describe the IP$_3$R dynamics, we followed Politi et al.'s derivation [30,33,34]. Thus, the dynamics of cytosolic Ca$^{2+}$ in a cell is given by:

$$\frac{dc_i}{dt} = J_{c_i} + \left[ k_1 \left( r_i \cdot \frac{c_i}{K_a + c_i} \frac{p_i}{K_p + p_i} \right)^3 + k_2 \right] (s_i - c_i) - V_{SERCA} \frac{c_i^2}{c_i^2 + K_{SERCA}^2},$$

(4)

where $k_1$ refers to maximal rate of Ca$^{2+}$ release, $K_a$ is the rate constant characterizing Ca$^{2+}$ binding to activating site in IP$_3$R, $K_p$ is the rate constant characterizing IP$_3$ binding to IP$_3$R, $k_2$ refers to Ca$^{2+}$ leak out of ER, $V_{SERCA}$ is the maximum rate of SERCA pump and $K_{SERCA}$ is the half activation constant. We assume that Ca$^{2+}$ acts as both a positive and negative regulator of IP$_3$R which is consistent with experimental observations of the single channel properties of the wild-type *Drosophila* receptor that has been studied using a lipid bilayer reconstitution technique [35]. Similar to IP$_3$, we also model diffusion of Ca$^{2+}$ through GJs by the following equation:

$$J_{c_i} \approx F_c \left[ \sum_{j \in N_i} c_j l_{ij} - c_j \left( \sum_{j \in N_j} l_{ij} \right) \right],$$

(5)

where $F_c$ refers to the permeability of Ca$^{2+}$ through GJs, $c_j$ refers to the concentration of Ca$^{2+}$ in neighboring cell $j$ and $l_{ij}$ refers to the length of cell boundary shared by cells $i$ and $j$ respectively.

Similarly, we describe the dynamics of Ca$^{2+}$ concentration in the ER of a cell as:

$$s_i(t) = \frac{c_{tot} - c_i(t)}{\beta},$$

(6)

where $s_i$ is the Ca$^{2+}$ concentration in the ER of the cell and $\beta$ is the ratio of effective cytoplasmic and effective ER volume, $c_i$ refers to the cytosolic Ca$^{2+}$ concentration in the cell and $c_{tot}$ refers to the total Ca$^{2+}$ concentration in the cell which includes both ER and the cytosol. We modified the rate of IP$_3$R inactivation term from the Politi model, $r$, to replicate our experimental

$Ca^{2+}$ dynamics. The modified equation is described below:

$$\frac{dr_i}{dt} = \frac{1}{\tau_{max}} \frac{k_\tau^4 + c_i^4}{k_\tau^4} \left(1 - r_i \frac{K_r + c_i}{K_r}\right)$$

A similar modification to the rate of inactivation term, $r$, has been proposed previously [36].

## IP$_3$R dynamics

We assume that the $Ca^{2+}$ binding to the inactivating site on the IP$_3$R is a slow process [36]. Consequently, we consider the dynamics of IP$_3$R inactivation by $Ca^{2+}$ in a cell as a separate differential equation given below:

$$\tau_r \frac{dr_i}{dt} = \left[1 - r_i \frac{(K_r + c_i)}{K_r}\right], \tag{7}$$

where $r_i$ refers to the fraction of IP$_3$Rs of cell $i$ that are not inactivated by $Ca^{2+}$, $K_r$ refers to the binding coefficient characterizing $Ca^{2+}$ binding to the inactive site on the IP$_3$R and $\tau_r$ refers to the characteristic time of IP$_3$R inactivation.

## Results

### The relative rate of IP$_3$ production governs transitions between classes of spatiotemporal $Ca^{2+}$ patterns at the tissue level

Multiple spatiotemporal classes of $Ca^{2+}$ activity are observed in vivo and ex vivo in the wing disc. However, an understanding of how this activity is regulated requires developing a systems-level description. To summarize, these include: (i) single-cell $Ca^{2+}$ spikes, (ii) intercellular $Ca^{2+}$ transients (ICTs), (iii) intercellular $Ca^{2+}$ waves (ICWs), and a (iv) global fluttering phenomenon (Fig 1C and Table 1 and S1–S8 Movies) [18–20]. The frequencies of these observed classes are dependent on the age of the larvae in both in vivo and ex vivo experiments. Younger larva with smaller discs (4–5 days after egg laying) exhibit higher occurrences of ICWs and fluttering states while older, larger larval discs (6–8 days after egg laying) predominantly show ICTs and spikes [19]. For ex vivo cultures, the transition from limited to tissue-wide $Ca^{2+}$ signaling activity depends on the amount of fly extract (FEX) added to the culture. Low concentrations of FEX stimulated $Ca^{2+}$ spikes. Progressively increasing levels of FEX resulted in ICTs, ICWs, and eventually fluttering. Further, FEX-stimulated $Ca^{2+}$ dynamics is based on IP$_3$R-based release of $Ca^{2+}$ from the ER to the cytosol as shown in Fig 1D [19].

These experimental findings motivated us to ask what specific cellular properties of the wing disc result in the emergence of these distinct patterns. To systematically investigate the underlying principles governing the emergence of these patterns, we formulated a two-dimensional image-based, geometrically realistic computational model of $Ca^{2+}$ signaling in the wing disc pouch where columnar epithelial cells are connected by gap junctions (S1 Fig), [37–40]. Image-based modelling enables the holistic characterization of molecular mechanisms and tissue dynamics during organogenesis [41]. Given the near universal conservation of the $Ca^{2+}$ signaling pathway across model systems [27,28], the baseline single-cell $Ca^{2+}$ model in our study was adapted from Politi and colleagues [30]. The model equations, biological relevance and descriptions of the parameters are shown in Fig 1D and Table 1.

To reproduce the four distinct patterns in silico, we varied the $V_{PLC}$ parameter progressively in individual cells across a range of values (Table 1). Given that our patterns were dependent on the concentration of FEX, we varied $V_{PLC}$ as a lumped-parameter representing the

**Table 1. Baseline parameters used in the model.**

| Parameter | Description | Value |
|---|---|---|
| $k_{5,P}$ | IP$_3$ dephosphorylation rate constant | 0.66 s$^{-1}$ |
| $K_{PLC}$ | Half activation of PLC | 0.2 μM |
| $V_{PLC}$ | Maximum production rate of PLC | 0.1–1.5 μM s$^{-1}$ |
| $\beta$ | Ratio of effective volumes ER/cytosol | 0.185 |
| $V_{SERCA}$ | Maximum SERCA pump rate | 0.9 μM s$^{-1}$ |
| $K_{SERCA}$ | Half activation constant for SERCA pump | 0.1 μM |
| $k_1$ | Maximum rate of Ca$^{2+}$ release from IP$_3$R | 1.11 s$^{-1}$ |
| $k_2$ | Ca$^{2+}$ leak from ER | 0.0203 s$^{-1}$ |
| $K_a$ | Ca$^{2+}$ binding to activating site of IP$_3$R | 0.08 μM |
| $K_r$ | Ca$^{2+}$ binding to inactivating site of IP$_3$R | 0.4 μM |
| $K_p$ | IP$_3$ binding to IP$_3$R | 0.13 μM |
| $\tau_{max}$ | Maximum time constant of IP$_3$R inactivation | 800 s$^{-1}$ |
| $k_\tau$ | Ca$^{2+}$ dependent rate of IP$_3$R inactivation | 1.5 μM |
| $F_p$ | GJ permeability for IP$_3$ | 0.005 μM$^2$ s$^{-1}$ |
| $F_c$ | GJ permeability for Ca$^{2+}$ | 0.0005 μM$^2$ s$^{-1}$ |
| $C_{tot}$ | Total Ca$^{2+}$ concentration in ER and cytosol | 2 μM |

Most baseline parameters were adopted from Politi and colleagues [30]. V$_{PLC}$ was varied in this report in order to investigate the effects of IP$_3$ production on spatiotemporal calcium patterns.

level of agonist stimulation (Fig 1D). We and others have demonstrated by FEX, which contains a mixture of agonists, stimulates PLC activity through GPCR and RTK signaling [18,19]. From these results, it can be inferred that the stimulation of PLC activity from FEX would subsequently increase the maximal rate of production of IP$_3$ (higher V$_{PLC}$). Thus, V$_{PLC}$ is not *directly* a parameter representing the concentration of FEX, but is a parameter that describes the net activity of PLC through FEX stimulation of upstream receptors. The computational model successfully reproduced the four different spatiotemporal classes of Ca$^{2+}$ signaling dynamics observed in vivo and ex vivo (Fig 2A–2D and S9–S12 Movies). Interestingly, we discovered that the formation of these patterns is dependent on the number of cells in the simulated tissue having a V$_{PLC}$ value below, above, or equal to the Hopf bifurcation threshold for single-cells (V$_{PLC}$ = 0.774) (Fig 2E). The Hopf threshold was identified from a single-cell version of the model wherein Ca$^{2+}$ oscillations occur in the cell when V$_{PLC}$ is at or above the value of 0.774 (Figs 2F and S2A and S3A). Simulated cells that have a V$_{PLC}$ value above the Hopf threshold, in the absence of agonist stimulation, are termed "initiator cells" and are posed to exhibit high levels of IP$_3$ production. Neighboring simulated cells with V$_{PLC}$ values below the Hopf threshold are termed "standby cells" that receive a signal from initiator cells to propagate a signal. For instance, if a majority of standby cells have V$_{PLC}$ values significantly below the critical Hopf bifurcation threshold (standby cell V$_{PLC}$ randomly uniformly distributed between 0.1–0.5), single-cell Ca$^{2+}$ spikes occur only where initiator cells oscillate (Fig 2A and 2E). When we increased standby cell V$_{PLC}$ values close to the lower end of the Hopf bifurcation point (Fig 2B and 2E), we noticed the formation of ICTs (standby cell V$_{PLC}$ randomly uniformly distributed between 0.25–0.60). Finally, we observed the formation of ICWs (standby cell V$_{PLC}$ randomly uniformly distributed between 0.4–0.8) and fluttering phenotypes (standby cell V$_{PLC}$ randomly uniformly distributed between 1.4–1.5) (Fig 2C–2E) for cases when the majority of cells in the system were assigned a V$_{PLC}$ close to or above the bifurcation threshold, thereby placing more cells in an initiator state. In the absence of initiator cells, Ca$^{2+}$ activity is

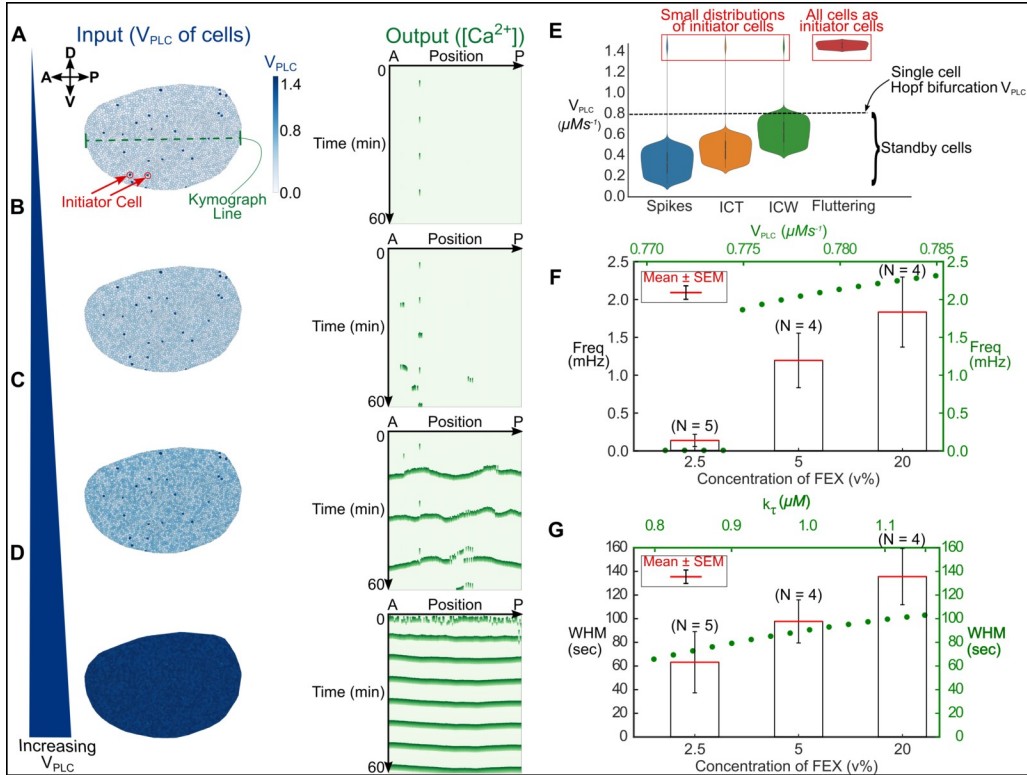

**Fig 2. The level of hormonal stimulation governs the spatial extent of intercellular Ca²⁺ communication. (A-D)** Computer simulations recapitulating the key classes of multicellular Ca²⁺ activity observed in vivo and ex vivo. **(A)** When the majority of cells have $V_{PLC}$ values below the Hopf bifurcation threshold (*left*), single-cell Ca²⁺ spikes are seen (*right*). Initiator cells (red arrows) are cells with $V_{PLC}$ values set between 1.4 and 1.5 in the simulation. A line through the A/P direction (green) demonstrates where the kymograph line is drawn that produces the simulated tissue's corresponding kymograph. **(B)** Intercellular Ca²⁺ transients are observed (*right*) as the distribution of $V_{PLC}$ in cells is increased (*left*). **(C)** A further increase in $V_{PLC}$ (*left*) results in the emergence of periodic intercellular Ca²⁺ waves (*right*). **(D)** "Fluttering" occurs (*right*) when $V_{PLC}$ levels in all of the cells in the disc is above Hopf bifurcation (*left*). **(E)** Quantification of $V_{PLC}$ distribution in the initiator and the standby cells for each of the classes of Ca²⁺ signaling activity. The first three Ca²⁺ signaling classes have a small distribution of initiator cells (red box) that are necessary for signal initiation. The dashed line indicates the threshold of the $V_{PLC}$ parameter that permits Ca²⁺ oscillations. **(F-G)** The single-cell version of our model predicts that the frequency and width at half maximum (WHM) of Ca²⁺ oscillations is altered by varying $V_{PLC}$ and $k_\tau$. This prediction matches the WHM of Ca²⁺ activity observed in ex vivo discs cultured with variable concentrations of fly extract. Error bars are reported as standard error of the means (SEM).

not observed. This suggests that initiator cells are necessary for the formation of Ca²⁺ transients in developing tissues.

The "single-cell" version of the model predicts that differences in Ca²⁺ signal amplitude and frequency are tunable by varying the $V_{PLC}$ and $k_\tau$ parameters (Figs 2F and 2G and S2). An output Ca²⁺ signal was observably tunable with $V_{PLC}$ variations in replication of the four distinct patterns (Fig 2A–2E). Because the single-cell model predicted Ca²⁺ signal perturbations by tuning $k_\tau$, we sought to investigate how the characteristic time associated with inactivation of IP₃R would influence tissue-scale signaling. To do this, we performed a sensitivity analysis of the two model parameters that influence characteristic time associated with inactivation of IP₃R: $k_\tau$ and $\tau_{max}$. Two-dimensional computational model simulations of the tissue were performed by varying parameter values as a percentage of their baseline values (Table 1) while holding other parameters constant. The baseline parameter set was selected such that the simulation generated intercellular Ca²⁺ waves (S4 Fig, red box). This was done by each baseline

simulation having the exact same $V_{PLC}$ profile for standby and initiator cells with standby cell $V_{PLC}$ values being uniformly random distributed between 0.7 and 1.0 and parameter values being set to those detailed in Table 1. Reducing $k_\tau$ leads to a narrower width at half maximum (WHM) of $Ca^{2+}$ transients (S4A Fig). A similar result is observed where a reduction of $\tau_{max}$ results in decreased WHM, whereas increased $\tau_{max}$ increased WHM (S4B Fig). Further, a decrease in the $\tau_{max}$ parameter increased the frequency of signals in the simulated tissue, while an increase reduced the frequency of $Ca^{2+}$ transients. This suggests that the system is more sensitive to $\tau_{max}$, and variations in $\tau_{max}$ have much greater impact in how quickly the system responds to stimulus.

These results indicate that tissue-level $Ca^{2+}$ patterns depend on the spatial distribution of cell states defined by their maximal $IP_3$ production rates, in relation to the effective tissue-level Hopf bifurcation threshold. Further, other key parameters in the model can be modified in a manner to allow tunable $Ca^{2+}$ signaling patterns. A systematic sensitivity analysis of other parameters in the model demonstrated perturbations to $Ca^{2+}$ signal strength, frequency, duration, and propagation (S4–S6 Figs). Further exploration and quantitative analysis of all parameters could allow an avenue to explore how tunable $Ca^{2+}$ signaling can induce a desired physiological outcome.

## GJ communication limits $Ca^{2+}$ spikes in the absence of hormonal stimulation

To elucidate how gap junction proteins alter $Ca^{2+}$ signals, we simulated a scenario where the initiator cells chosen at random had their $V_{PLC}$ values set to the Hopf bifurcation threshold value of 0.774 and standby cells had a $V_{PLC}$ values that were randomly uniformly distributed between 0.1 and 0.5 (Fig 3A). Under these conditions, no $Ca^{2+}$ activity was observed in the presence of normal functioning GJ communication (Fig 3A'). Next, we compared the effect of blocking gap junction communication in silico. To do so, we set the permeability terms for $Ca^{2+}$ ($F_c$) and for $IP_3$ ($F_p$) to zero. Strikingly, we observed $Ca^{2+}$ spikes in simulated wing disc cells in the absence of GJ communication (Fig 3A"). We explored this phenomenon computationally by considering a single stimulated cell connected to neighboring cells by GJ communication by performing bifurcation analysis on our modified model for a single cell. We observed the emergence of a Hopf bifurcation as expected (S3A Fig). Next, the effect of the initial Hopf bifurcation point ($HB_1$) on gap junctional (GJ) permeability of $IP_3$, $F_p$, was analyzed. Setting $F_c$ to zero and progressively increasing $F_p$ increased the critical maximum rate of $IP_3$ activation threshold $V_{PLC}^*$ where $HB_1$ occurred (S3B Fig). Similar trends were observed when $F_c$ was increased to 0.05. Thus, our model suggests that inhibition of GJ communication lowers the Hopf threshold necessary to generate $Ca^{2+}$ activity in wing disc epithelial cells.

A closer look into the importance of GJ permeability on the formation of $Ca^{2+}$ signals was taken by varying $F_c$ and $F_p$ in computational simulations. Similar to the sensitivity analysis performed on $k_\tau$ and $\tau_{max}$, GJ permeability was varied by fixed percentages (S4C Fig). We discovered that decreased GJ permeability decreased the propagation of the $Ca^{2+}$ signal across the simulated tissue. However, the fixed percentage values ranging from 50% to 150% of the baseline parameter values (Table 1) did not produce notable changes of the ICW $Ca^{2+}$ pattern. To further investigate this, a wider range of fixed percentages were tested between 1% and 1000% of the baseline parameter values (S7A Fig). Starting from a baseline ICW, a 90% decrease in GJ permeability resulted in a transition from ICWs to ICTs, and eventually to single-cell $Ca^{2+}$ spikes, while a 100% increase in GJ permeability increased $Ca^{2+}$ signal propagation and decreased the frequency (S7A Fig). These findings show that GJ permeability alters the cytosolic residence time for critical messengers such as $IP_3$ and $Ca^{2+}$ whose cytosolic

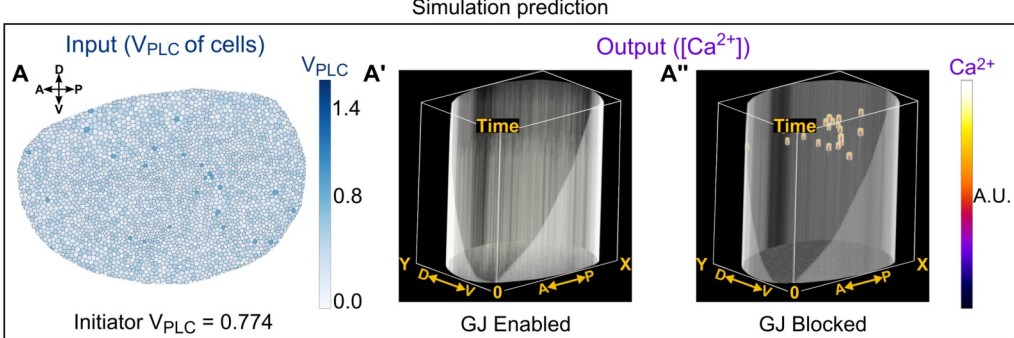

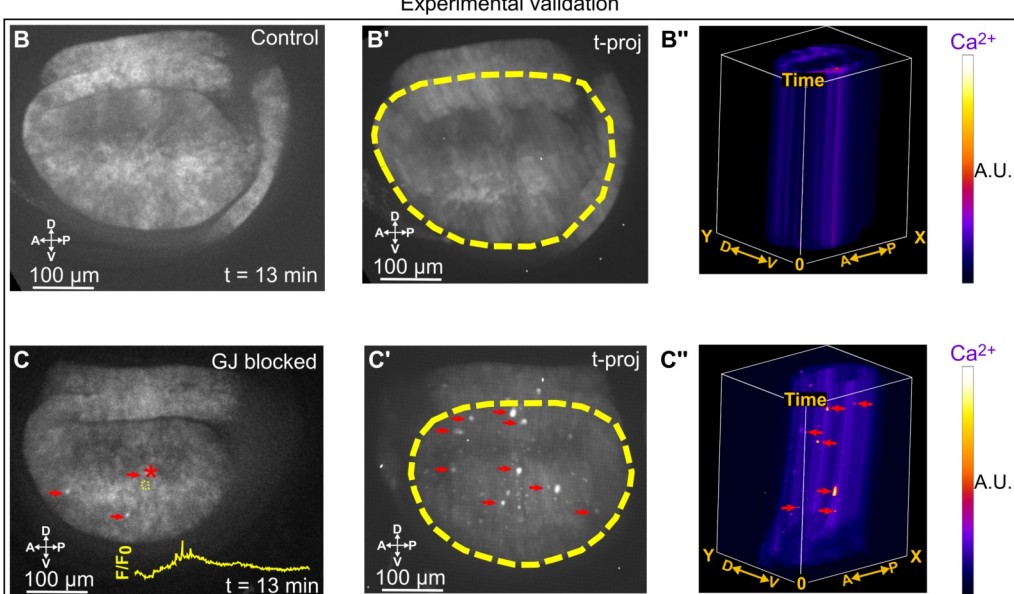

**Fig 3. Gap junction (GJ) communication decreases the proportion of cells exhibiting Ca²⁺ spikes. (A)** Simulation of Ca²⁺ signaling in wing disc where the $V_{PLC}$ values of initiator cells were set to the Hopf bifurcation threshold 0.774, and standby cell $V_{PLC}$ values were randomly distributed between 0.1 and 0.5. **(A')** Allowing GJ communication by letting permeability of IP₃ ($F_p$) and Ca²⁺ ($F_c$) be 0.005 μM² s⁻¹ and 0.0005 μM² s⁻¹ respectively, resulted in no Ca²⁺ activity (GJ Enabled). **(A'')** $F_p$ and $F_c$ were set to zero to simulate inhibition of GJ communication (GJ Blocked). Inhibition of GJ communication resulted in Ca²⁺ spike activity. A.U. indicates arbitrary units that correspond to the intensity of the signal. X coordinates roughly correspond to the A/P direction of the wing disc pouch whereas the Y coordinate principally describes the D/V axis of the pouch. **(B)** Ex vivo time lapses of nub>GCaMP6f x UAS-RyRᴿᴺᴬⁱ (control) wing discs in Grace's low ecdysone media were generated by imaging for 1 h at 10 sec intervals. Under this condition, we observed no Ca²⁺ activity in the wing disc cells. **(B')** Time-projection of the time lapse stack. The wing disc boundary is indicated with the yellow dashed line. **(B'')** A kymograph generated further demonstrates no instances of Ca²⁺ activity. **(C)** GJ communication was blocked by culturing wing discs in Grace's low ecdysone media with 100 mM of Carbenoxolone (CBX). Instances of spike activity are denoted by red arrows. The intensity of a region of interest (yellow dashed circle) is overlaid to demonstrate a spike in local Ca²⁺ activity. Ca²⁺ spike is observed when the intensity normalized to basal intensity is plotted (yellow line, $F/F_0$). **(C')** Time-projection of the time lapse movies. We observed a significant number of Ca²⁺ spikes in the 1 h time interval when GJs were inhibited. Yellow dashed lines indicated disc boundary. **(C'')** A kymograph generated demonstrates instances of Ca²⁺ spike activity.

concentrations affects Ca²⁺ release from ER in both initiator cells and standby cells. This is consistent with bifurcation analysis demonstrating that GJ permeability influences stimulation threshold required for Ca²⁺ oscillations in cells (S3B Fig).

We tested these computational modeling predictions experimentally. To do so, we pharmacologically inhibited the gap junctional protein Inx2 using Carbenoxolone (CBX) and

observed the emergence of $Ca^{2+}$ spikes in the absence of FEX in the culture media (Fig 3B-3B" and 3C-3C", and S13 and S14 Movies). This further demonstrates that gap junction communication regulates $Ca^{2+}$ dynamics in the wing disc pouch.

## GJ permeability modulates $Ca^{2+}$ signaling during development

Of note, the integrated intensity of $Ca^{2+}$ signaling throughout the wing disc pouch decreases during development suggesting an inverse relationship between $Ca^{2+}$ signaling activity and organ growth rates [19]. Therefore, we investigated the role of tissue size in altering $Ca^{2+}$ signaling dynamics to propose an explanation for this finding. To do so, we simulated $Ca^{2+}$ signaling in different sized wing discs. We hypothesized two different possible scenarios that could explain a decrease in integrated $Ca^{2+}$ signaling activity. In the first scenario, the total number initiator cells was allowed to decrease in a power law fashion as development proceeds while GJ permeability remained constant. This resulted in a decay in total $Ca^{2+}$ signaling activity with a transition from intercellular waves to predominantly single-cell spiking activity (Fig 4A). The fraction of initiator cells was varied according to the following equation:

$$N_{initiators} = 8,000 \cdot N_{cells}^{-0.8}, \tag{8}$$

where $N_{cells}$ is the total number of cells in the simulated tissue. The power-law relationship exponent value of -0.8 was used from the discovery that integrated $Ca^{2+}$ intensity scales in a similar power-law fashion detailed in one of our previous publications [19]. The constant in front of the equation, 8,000, was chosen to output physiologically realistic fractions of initiator cells with the simulated tissue sizes. In the second scenario, GJ permeability was set to decrease with increasing organ size while the total number of initiator cells was held constant at $N_{initiators}$ = 65 (Fig 4B). GJ permeability was varied according to the following equation:

$$F_p = 800 \cdot N_{cells}^{-1.8}, \tag{9}$$

where $F_c$ is directly proportional to $F_p$ such that $F_c = 0.1 \cdot F_p$. Similar to the scenario where initiator cell count was scaled, we investigated a scenario of scaling gap junction communication. We selected a power-law relationship value of -1.8 as an analogy to the relationship between integrated $Ca^{2+}$ signaling activity we reported previously [19]. To ensure consistency across the two simulation setups, all other parameter values aside from initiator cell count and GJ permeability are those listed in Table 1 with $V_{PLC}$ values of standby cell equal to 0.40. Simulations corresponding to each scenario show a decrease in progression of $Ca^{2+}$ signaling activity starting from ICWs and intercellular transients in smaller simulated discs and decaying to intermittent single-cell spikes in larger simulated discs (Fig 4A and 4B). This suggests that both scenarios provide a possible explanation for the decrease in integrated $Ca^{2+}$ signaling activity observed in wing discs as development progresses.

To distinguish between these scenarios, we cultured wing discs from multiple developmental stages from days 5–7 after egg laying (AEL) without any agonist stimulation and observed the resulting $Ca^{2+}$ dynamics. We reasoned that the lack of agonist stimulation would reveal cells with phospholipase activity sufficient to create spikes, and this could further be considered initiator cells. We observed single-cell $Ca^{2+}$ spikes that we interpret as characteristic of "initiator" cells in all samples independent of sizes. We did not observe $Ca^{2+}$ transients or waves in unstimulated smaller wing discs obtained day 5 AEL. This is consistent with agonist stimulation increase in PLC activity in standby cells. We next characterized the total number of spikes in the discs of all sizes. We found a positive correlation between the total number of spikes and the size of the disc pouch (Fig 4C). The difference in spiking activity between discs of varying ages was not significant when scaled for pouch size (Fig 4D). This is consistent with

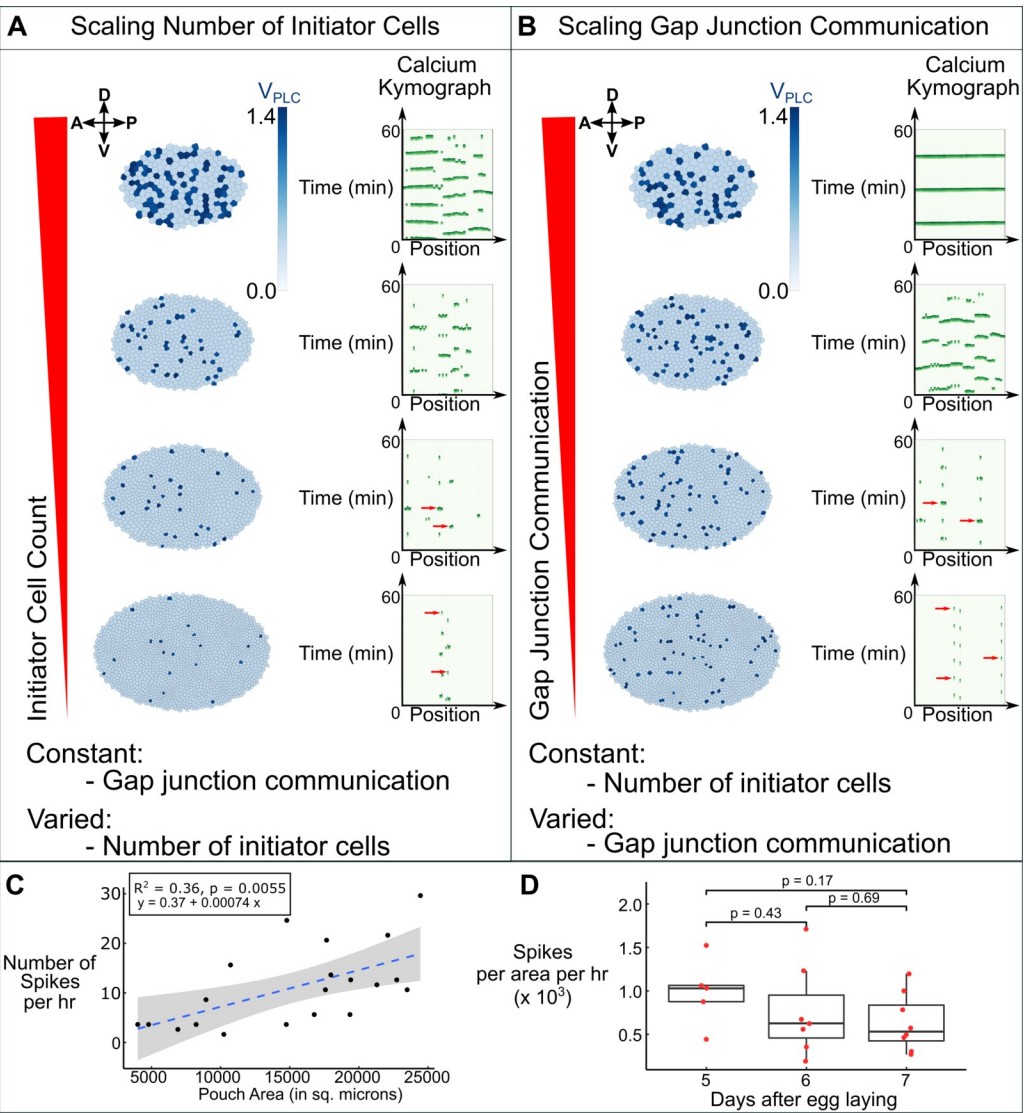

**Fig 4. GJ permeability determines total tissue-level signaling activity as development progresses. (A-B)** Simulations of $Ca^{2+}$ signaling for wing discs of increasing size. **(A)** *(Left column)* The total number of initiator cells was varied with tissue size according to the following equation: $N_{initiators} = 8,000 \cdot N_{cells}^{-0.8}$ while GJ permeability was held constant. The $V_{PLC}$ of all standby cells was restricted to values of 0.40, with initiator cells being denoted by dark blue cells with $V_{PLC}$ values between 1.3 and 1.5. *(Right column)* Associated 2D kymographs of the simulated pouches shown in **A**. **(B)** *(Left column)* The GJ permeability is varied while holding the total number of initiator cells constant. GJ permeability was varied according to the following equation: $F_p = 800 \cdot N_{cells}^{-1.8}$ and $F_c = 0.1 \cdot F_p$. The total number of initiator cells were held constant in this scenario ($N_{initiator} = 65$) and standby cells were restricted to $V_{PLC}$ values of 0.40. *(Right column)* Associated kymographs of the simulated pouches shown in **B**. Both scaling models demonstrate high $Ca^{2+}$ signaling activity in small discs and gradually regress to low $Ca^{2+}$ signaling activity in large discs. **(C-D)** Experimental validation of the computational predictions in which the discs were cultured ex vivo in Grace's low 20E media (basal media) for 1 h in the absence of any stimulus triggering $Ca^{2+}$. **(C)** Quantification of total number of $Ca^{2+}$ spikes in different sized wing disc pouch during 1 h culture. A linear regression line was fit to the data set and the p-value for the slope of the fitted line is shown. Since the *p*-value is less than 0.05 (level of significance), a positive correlation between size and the number of spikes could be inferred. Grey region corresponds to 95% confidence bands of the trend line. **(D)** Quantification of total number of spikes in disc pouch scaled with respect to pouch size during various stages of larval development. *p*-values were obtained by Mann Whitney U test.

a scenario of a relatively constant number of initiator cells in the system with overall GJ permeability decreasing as the organ reaches its terminal size. This scenario is further supported by findings from previous reports showing a decrease in GJ permeability as larval development proceeds [39,40]. Furthermore, a decrease in GJ permeability increases the net cytosolic residence time and effective concentrations of $IP_3$ and $Ca^{2+}$ within the cytosol leading to an increased instance of cytosolic $Ca^{2+}$ spikes.

## Gαq overexpression induces intercellular $Ca^{2+}$ waves and reduces wing size

Next, using the GAL4/UAS system (S8 Fig), we overexpressed the wild type splice 3 variant of *Drosophila* Gαq in the wing disc to characterize how different classes of upstream signals impact the spatiotemporal patterns of $Ca^{2+}$ signaling [42]. G protein-coupled receptor (GPCR) activation stimulates Gαq-driven PLC$\beta$ activity to generate $IP_3$ and $Ca^{2+}$ [43]. Strikingly, ectopic Gαq expression was sufficient to generate robust formation of intercellular waves independent of the presence of FEX in the media (Fig 5D and 5D'). The waves were periodic in nature and were similar to FEX-induced waves (Fig 1C). This experimental finding most likely resembles our previous simulation of ICWs with a small fraction of randomly located initiator cells surrounded by standby cells (Fig 2C). Additionally, the wing disc size (day 6 AEL) and adult wing size were significantly reduced when Gαq was overexpressed (Fig 5E and 5F). To understand whether the reduction in the wing size was due to changes in proliferation or cell growth, we quantified the total number of cells in the region bounded by the LIII, LIV, ACV wing veins and the wing margin. We observed a reduction in the total number of trichomes, where each individual trichome corresponds to a cell [44]. Furthermore, we found that cell size was reduced when Gαq was overexpressed (S9F Fig). However, the wing shape is not significantly affected when Gαq is overexpressed (S9G Fig). In sum, increasing the concentration of Gαq in the pouch is sufficient to generate periodic $Ca^{2+}$ waves. Further, these periodic $Ca^{2+}$ waves are correlated with reduction in wing and disc size suggesting that tissue-wide $Ca^{2+}$ wave activity may play a role in determining final organ size via growth inhibition.

## Insulin signaling increases wing size but only generates localized $Ca^{2+}$ signals

Because FEX is an undefined cocktail of biochemical factors, we tested whether specific ligands added to the organ culture affects $Ca^{2+}$ signaling activity and growth. In addition to FEX, insulin is often added to organ culture media to stimulate cell proliferation [45,46]. Hence, we tested whether insulin signaling regulates $Ca^{2+}$ signaling activity independent of FEX. Similar to other experiments, we upregulated and downregulated insulin signaling in the wing disc using the GAL4/UAS expression system (S8 Fig). As expected, wing disc size and adult wing size were decreased when insulin signaling is downregulated (Fig 5C", 5E, and 5F). Strikingly, we observed that activation of insulin stimulated pathways results in localized $Ca^{2+}$ spikes and ICTs (Fig 5B and 5B'). Titrated concentration of insulin in the culture media demonstrated that a higher concentration of insulin increased the number of spikes (Fig 5G). Quantification of the $Ca^{2+}$ spikes showed a positive correlation between spikes normalized to area of the pouch with log of insulin concentration (Fig 5G). However, even high concentrations of insulin were not sufficient to generate periodic ICWs. In contrast, expressing a dominant negative form of the insulin receptor resulted in minimal $Ca^{2+}$ spiking activity (Fig 5C and 5C') [47]. These results demonstrate that insulin signaling stimulates $Ca^{2+}$ activity in the wing disc. Overall, these results indicate that the agonist class encodes different spatiotemporal dynamics of $Ca^{2+}$ signaling at the tissue scale.

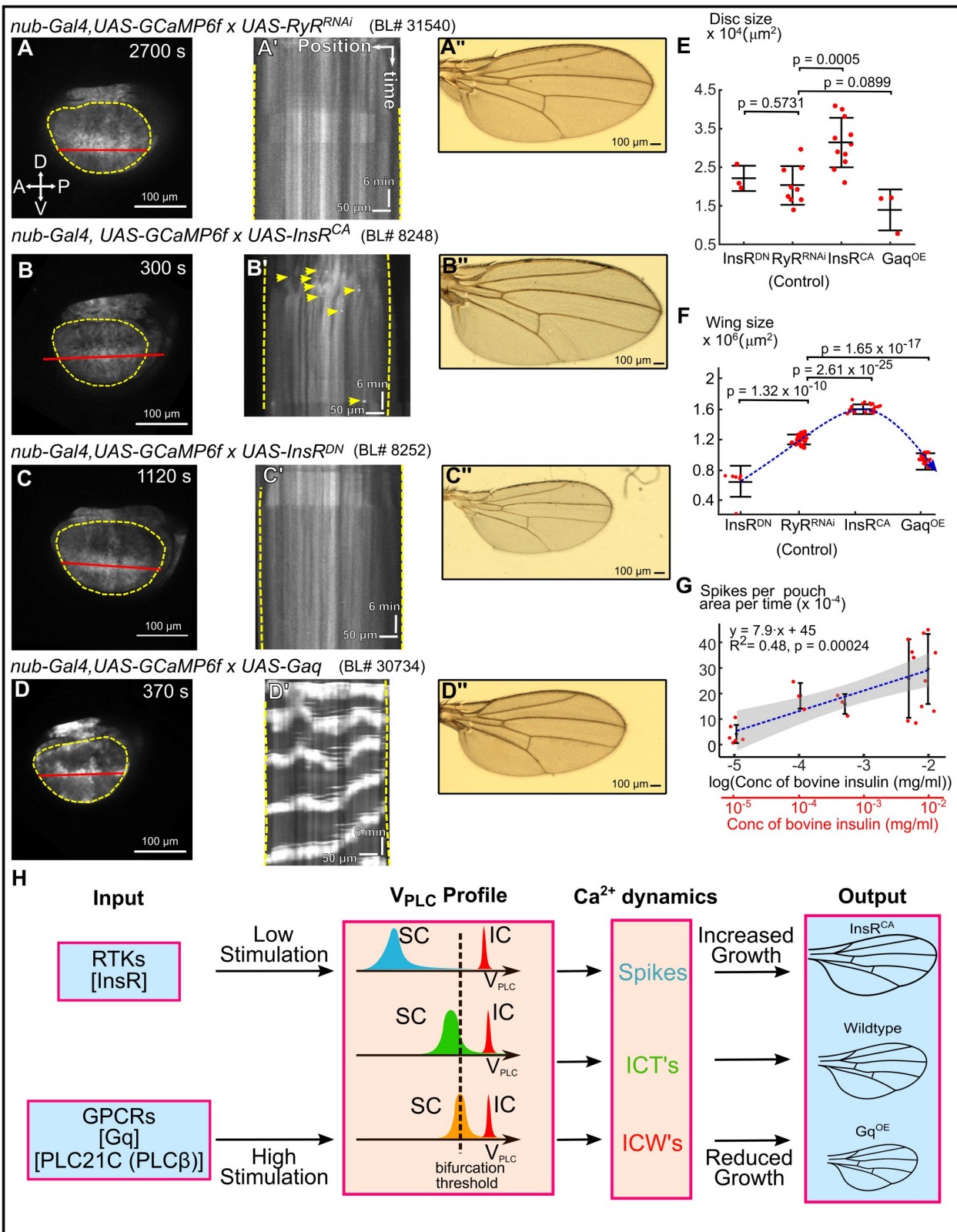

**Fig 5. GPCR and insulin signaling regulate wing size and differentially regulate Ca²⁺ signaling.** (A-D) Montages of time-lapse movies of wing discs cultured ex vivo. (A'-D') Kymographs of the corresponding time-lapse movies. (A"-D") Adult wings from the indicated genetic perturbation. (E) Quantification of the wing disc sizes for the indicated genetic perturbations. (F) Quantification of the adult wing sizes for the indicated

perturbations. **(G)** Quantification of $Ca^{2+}$ spikes when insulin dose is progressively increased in ex vivo cultures. A linear regression trend line was fit to the data and the *p*-value of the slope is shown. Since the *p*-value is less than 0.05, a positive correlation between spikes and the log (concentration) can be inferred. Grey region illustrates 95% confidence bands of the linear regression **(H)** Summary of key findings based on the proposed model for tissue-level regulation of dynamics in epithelial tissues. Different cell surface receptor stimulation produces varying $V_{PLC}$ profiles of developing tissues causing $Ca^{2+}$ signaling and varied tissue responses. Scale bars in (A-D) and (A"-D") represent 100 μm, while yellow dotted lines indicate pouch boundaries, and red lines indicate x-y positions in the kymograph. Horizontal scale bars in (A'-D') represent 50 μm. Vertical scale bars in (A'-D') represent 6 min. Student t-test was performed. Labels in (E) represent crosses of UAS-transgene to parental nub>GCaMP6f in the case of $InsR^{CA}$ and $InsR^{DN}$ or nub>GCaMP6f; mcherry in the case of $RyR^{RNAi}$ or Gαq. The UAS lines used are UAS-RyR$^{RNAi}$ (BL#31540), UAS InsR$^{CA}$ (BL#8248), UAS-InsR$^{DN}$ (BL#8252) and UAS-Gq (BL#30734) respectively.

In silico simulations with parameter variations in half activation of the PLC parameter, $K_{PLC}$, and GJ permeability were performed to test scenarios of how insulin signaling can only generate $Ca^{2+}$ spiking activity even under saturating conditions (S7B and S10 Figs). These two parameters were chosen because $K_{PLC}$ can potentially be influenced by downstream of insulin receptor activity, a receptor tyrosine kinase (Fig 1D) or by the biochemical activity of phospholipase C γ. Additionally, single-cell spikes were only observed when gap junction communication was inhibited either experimentally (Fig 3B and 3C) or in silico (Figs 3A and S7A). Although large $K_{PLC}$ variations influenced signal frequency (S7B Fig), there was no instance in which varying $K_{PLC}$ resulted in exclusive production of $Ca^{2+}$ spikes. To further investigate this, both GJ permeability and $K_{PLC}$ were varied simultaneously (S10 Fig). From a baseline intercellular wave (S10A Fig), $K_{PLC}$ was increased from its baseline value while GJ permeability was simultaneously decreased from its baseline value. Only when GJ permeability was decreased was there observance of single-cell $Ca^{2+}$ spikes (S10 Fig). Thus, the computational model suggests that insulin signaling must not only stimulate $IP_3$ production, but also inhibit GJ permeability to account for the limitations of spatial spread of $Ca^{2+}$ signaling.

As a first step to assess whether $Ca^{2+}$ levels in the cell directly control organ sizes, we exploited the $Ca^{2+}$ buffering effects induced by high levels of GCaMP6f sensor expression. The GCaMP6f sensor consists of a M13 fragment of myosin light chain kinase, GFP and Calmodulin (CaM) to which $Ca^{2+}$ binds [48]. To elucidate the role of $Ca^{2+}$ signaling in insulin mediated growth, we compared the effects of co-expressing the GCaMP6f sensor ($K_d = 375\pm14$ *nM*) to decrease the amount of free cytosolic $Ca^{2+}$ [49]. We found that co-expressing GCaMP6f sensor increased wing size when insulin signaling was also stimulated (S11 Fig). This increase in size was also observed in control wing disc without insulin upregulation (S11A and S11B Fig). This analysis provides additional evidence that buffering cytosolic $Ca^{2+}$ signaling influences the final tissue size. In contrast, when we expressed Gαq and GCaMP6f, we did not observe a significant decrease in size when compared to just expressed Gαq without GCaMP6f (S11C and S11D Fig). This may be due to the inability of GCaMP6f expression to buffer the high levels of $Ca^{2+}$ in the cytosol when Gαq is overexpressed. We also observed a severe reduction in the wing size, along with vein defects, in flies that were homozygous for the GCaMP6f sensor, consistent with a buffering role for high concentrations of GCaMP6f expression (S12 Fig). To validate this finding, we compared the adult wing sizes from flies expressed one and two copies of GCaMP6f. Strikingly, increasing the dose of the relative GCaM6f expression decreased the size (S12C and S12D Fig). We observed a severe reduction in size as expression of the transgene was further increased with the presence of two copies of the GAL4 driver. To further elucidate the sponging effects of $Ca^{2+}$, we expressed CaMKII, which binds to $Ca^{2+}$/CaM complex [50]. Similar to overexpressing GCaM6f, we observe a significant increase in the wing area (S12E and S12F Fig). Collectively, these results support a role of cytosolic $Ca^{2+}$ concentration as either a growth enhancer or suppressor and that an optimal amount of $Ca^{2+}$ signaling is required for robust size control.

## Discussion

The main finding of this work is the discovery of a parsimonious mechanistic model that links global hormonal stimulation of $Ca^{2+}$ signaling to emergent spatiotemporal classes of signaling dynamics. Further, we identify downstream correlations to the final organ size that suggest these signaling dynamics may mediate growth-related information used by the system to tune organ size. To do so, we developed a geometrically accurate computational model of $Ca^{2+}$ signaling in the *Drosophila* wing imaginal disc, a premier model for studying conserved cell signaling mechanisms [23,51,52]. Previously, we discovered a correlation between $Ca^{2+}$ signaling during larval growth and final organ size. Four distinct patterns of $Ca^{2+}$ signaling activity occur in the wing imaginal disc pouch as observed in vivo and ex vivo experiments [19]. Through systems-level computational analysis, we established the essential conditions required for generating the different patterns.

The model predicts two cell types with different levels of $IP_3$ production: "initiator cells" with high $IP_3$ production and "standby cells" with baseline $IP_3$ production levels. The presence of initiator cells is necessary, but not sufficient, to induce multicellular $Ca^{2+}$ signaling. Additionally, the distribution of the maximal rate of $IP_3$ production, $V_{PLC}$, in the standby cells determines the spatial range of $Ca^{2+}$ signaling. As $V_{PLC}$ values of standby cells approaches the Hopf bifurcation threshold, $Ca^{2+}$ activity transitions from single-cell signals toward global signals.

What are the possible functional implications of a tissue consisting of initiator and standby cells? A recent study on electrical signal transmission in bacterial communities suggests that the transition from localized short-range signaling to global community-level communication is associated with a cost-benefit balance [53]. In that context, long-range signaling increases the overall fitness of the community against chemical attacks, while a reduction in growth rate is the cost to individual cells. In the wing disc, a similar analogy can be drawn where significant generation of long-range $Ca^{2+}$ signals due to overexpression of Gαq results in reduced wing disc growth. Thus, the proposed model in this work can also be characterized as a cost-benefit tradeoff within the context of tissue-level signaling. For instance, it has been suggested that the fast $Ca^{2+}$ waves facilitate migration and proliferation of the healing cells by inhibiting excessive apoptotic response during wound healing in epithelia [54].

Our model also predicts that the inhibition of GJs lowers the Hopf threshold necessary for generating $Ca^{2+}$ spikes. We have validated this prediction experimentally where inhibition of GJs results in the formation of $Ca^{2+}$ spikes in the absence of external agonist. Further, computational simulations demonstrate that as GJ permeability decreases, there is a transition of activity from synchronous global to asynchronous local $Ca^{2+}$ activity. How gap junctional mediated $Ca^{2+}$ signaling is connected to the regulation of cell mechanics is not currently understood and warrants further investigation. One possibility is that tension impacts the level of gap junction communication between cell and may also influence the activity of mechanosensitive ion channels [55]. Feedback between $Ca^{2+}$ signaling and cell mechanics may play important roles in ensuring tissue growth and morphogenesis. This is evident from our previous experimental work and other experimental studies in the literature where knockdown of gap junctional proteins such as Inx2 leads to a reduction in wing and eye size in *Drosophila* [19,56].

Upregulated insulin signaling increases the formation of $Ca^{2+}$ spikes. One possible implication is that insulin signaling inhibits GJs in addition to increasing PLCγ activity. This implication is consistent with the role of insulin signaling in inhibiting gap junction proteins Connexin43 in vertebrates [57–59]. Thus, the increase of adult wing and developing wing disc size from higher insulin activity correlates with higher levels of localized $Ca^{2+}$ spiking activity,

which has a limited total integrated calcium signal at that tissue level. In contrast, $Ca^{2+}$ waves induced by Gαq overexpression are correlated with smaller adult wings and wing discs. Higher Gαq expression, which activates PLCβ activity results in robust production of tissue-scale intercellular $Ca^{2+}$ waves. We show experimentally that insulin signaling controls $Ca^{2+}$ spike activity in the wing disc, potentially through GJ inhibition or PLCγ activation, whereas GPCR-based Gαq signaling is sufficient to generate global $Ca^{2+}$ waves. These results are consistent with previous reports of $Ca^{2+}$ spikes being observed in discs that have reached their final size, and $Ca^{2+}$ waves being observed in smaller developing discs [19]. A recent study found that $Ca^{2+}$ signals are initiated in response to wounding by the G-protein coupled receptor Methuselah like 10 (Mthl10) [60]. Mthl10 is activated by Growth-blocking peptides (Gbps) [60]. Whether Mthl10 also is involved in developmental growth requires further investigation, but may be consistent with our findings that intercellular calcium wave activity inhibits organ growth. These findings suggest that $Ca^{2+}$ acts as both a growth enhancing and growth inhibiting signal dependent upon the tissue-level scale of the activity and level of gap junction coupling.

The spatial range of tissue-level signaling is determined by how the $IP_3$ production is organized with respect to a Hopf bifurcation threshold throughout the tissue. Localized transients are correlated with larger wings induced by insulin-stimulated growth whereas global signaling is correlated with smaller wings that are stimulated by upstream GPCRs and Gαq upregulation (Fig 5H). This resembles a paradoxical network motif [61] where $Ca^{2+}$ signaling has two opposite effects on the same downstream target, dependent upon the tissue-level magnitude of the $Ca^{2+}$ signaling. Within the context of the hypothesized $IP_3$/shunt model proposed in our previous study, the strong induction of $Ca^{2+}$ waves will reduce the level of phosphatidylinositol 4,5-bisphosphate ($PIP_2$), a key substrate for growth [19]. This may occur as high levels of Gαq/PLC activity are proposed to deplete $PIP_2$ levels [62]. This would occur due to substrate depletion of $PIP_2$ through promotion of $IP_3$ generation and downstream activity by stimulation of PLC activity. In turn, this would lead to reduced availability of $PIP_2$ for conversion of $PIP_2$ to phosphatidylinositol-trisphosphate ($PIP_3$), a key second messenger for stimulating protein kinase AKT and downstream growth promotion [63].

Similar to the reduced wing size observed in this study due to overexpression of Gαq, we have also reported that knockdown of Gαq gene decreases wing size in our previous study (S2 Table) [19]. Comparing the reduction in wing size due to perturbation of $Ca^{2+}$ signaling components with known size control genes such as morphogens (Dpp, Wg, Hippo) and mechanical transducers (RoK) indicate that the reduction in size is comparable to when $Ca^{2+}$ signaling is perturbed (S2 Table). Taken together, these experimental findings imply that Gαq signaling is paradoxical in nature. In the context of conserved network motif observed in biological systems, paradoxical components have the ability to activate and inhibit the downstream target despite a single source of stimulus [61]. Gαq could possibly be a growth promoter and growth inhibitor. This correlation motivates further studies that map out the exact molecular players that are downstream of Gαq signaling. This will require careful quantification of $PIP_2$ and $PIP_3$ under genetic perturbations of GPCR signaling. Additionally, future work is needed to quantify the metabolic benefits and costs of $Ca^{2+}$ signaling during tissue growth to observe if abundant use of metabolic resources to consistently propagate global activity is explanatory for the reduction in size in Gαq overexpression wings.

## Materials and methods

### Drosophila genetics

We used the GAL4/UAS system to express modulators of the $Ca^{2+}$ signaling pathway in the wing disc [64,65]. A nub-GAL4, UAS-GCaMP6f reporter tester line was created by

recombining nub-GAL4 and UAS-GCaMP6f lines [66]. Additionally, a second tester line was used that also includes UAS-mcherry. Gene perturbations were generated by crossing the tester line to either RNAi-based transgenic lines (UAS-Gene X$^{RNAi}$) or gene overexpression (UAS-Gene X). The following UAS transgenic lines were used: UAS-RyR$^{RNAi}$ (BL#31540) [67], UAS-Gq (BL#30734) [67], UAS-InsR$^{CA}$ (BL#8248) [68], UAS-InsR$^{DN}$ (BL#8252) [69]. Progeny wing phenotypes are from F1 male progeny emerging form the nub-Gal4, UAS-GCaMP6f/CyO x UAS-X cross or nub-Gal4, UAS-GCaMP6f/CyO; UAS-mcherry x UAS-X cross. Flies were raised at 25˚C and on a 12-hour light cycle.

## Live imaging

Wandering third instar larva approximately 6 days after egg laying were dissected in ZB media with 15% fly extract to obtain wing discs [19]. ZB media + 15% fly extract contains 79.4% (v/v) ZB media, 0.6% (v/v) of 1 mg/ml of insulin (Sigma aldrich), 15% ZB-based fly extract and 5% penicillin/streptomycin (Gibco). Wing discs were loaded into the previously described REM-Chip [70] and imaged using Nikon Eclipse Ti confocal microscope with a Yokogawa spinning disc and MicroPoint laser ablation system. Image data were collected on an IXonEM +colled CCD camera (Andor technology, South Windsor, CT) using MetaMorph v7.7.9 software (Molecular devices, Sunnyvale, CA). Discs were imaged at three z-planes with a step size of 10 μm, 20x magnification and 10-seconds intervals for a total period of one hour, with 200 ms exposure time, and 50 nW, 488 nm laser exposure at 44% laser intensity. We blocked GJ by inhibiting innexin-2 using Carbenoxolone (Cbx, Sigma Aldrich) drug [66]. Wing discs were incubated in ZB + 15% FEX with 30 μM Cbx for one hour before imaging. To induce Ca$^{2+}$ transients, we imaged wing discs in ZB media + 2.5% FEX [71]. Ca$^{2+}$ waves were induced by imaging the wing disc in ZB media + 15% FEX. Ca$^{2+}$ fluttering was observed when discs were imaged in ZB media + 40% FEX respectively. For experiments reported in Figs 3, 4, and 5, wing imaginal discs were cultured in Grace's media with low 20E (Dye et al., 2017). Briefly, basal Grace's media was prepared by addition of Grace's medium (Sigma, G9771) with 5 mM BisTris, 5% fetal bovine serium (FBS; ThermoFisher/Invitrogen, 10370098) and Pennicillin-Streptomyocin (Sigma P4333, 100x stock solution) along with 20 nM 20E (Sigma, H5142). For the Cbx experiment reported in Fig 3, we cultured wing discs in Grace's cocktail with 30 uM Cbx (Sigma Aldrich). For insulin dose response experiments reported in Fig 5, we added appropriate volume of Bovine insulin (Sigma, I5500) to the Grace's cocktail respectively.

## Image processing

All the images were processing in FIJI. Volume viewer plugin was used to generate 3D Kymographs. Briefly, TIFF stacks had background subtracted using the rolling ball background subtraction algorithm with a rolling ball radius of 15. The TIFF stack was then processed by the volume viewer plugin. Stacks were adjusted using the base Fire LUT setting to portray the signaling intensities. A similar approach was followed for the simulation outputs.

## Model formulation

Fig 1 summarizes the experimental system and data. Different classes of patterns emerge at the tissue-level as the level of global stimulation increases: spikes, intercellular Ca$^{2+}$ transients (ICTs), intercellular Ca$^{2+}$ waves (ICWs) and global fluttering [19]. However, a mechanistic understanding linking hormonal stimulation levels to transitions in these qualitative classes of organ-level signaling is lacking. We therefore formulated a computational model to test mechanistic hypotheses that could explain the observed Ca$^{2+}$ signaling dynamics.

### Intracellular model

A modified model of $Ca^{2+}$ signaling toolkit is based on adaptation of previous single-cell model of calcium signaling [30]. The model is summarized in the computational model section of the main text. To recapitulate the same time resolution as the experiments, the simulation time is 1 hour and for generating movies, samples are obtained every 10 s.

### Tissue model

For constructing a realistic model of the tissue, we used experimental images of a wing pouch to build an accurate model of the tissue structure. S1 Fig depicts the structure of the tissue used for simulations and the statistics of the corresponding network. A pouch was constructed computationally using EpiTools. We first segmented the apical cell boundaries using images from Ecad::GFP line. Then, the centroids of segmented cells were used to define cellular positions in the simulated wing disc. A Voronoi tessellation followed by multiple rounds of Lloyd's relaxation was used to define a template wing disc that matches the experimentally observed network topology.

### Quantification and statistical analysis

**Quantification of adult wings and statistics.** Total wing area was measured using ImageJ. We traced the wing margin by following veins L1 and L5 and the wing hinge region was excluded from the size analysis. All statistical analyses were performed using MATLAB or R. For comparisons, we used student t-tests to assess the statistical significance. *p*-value, standard deviation and sample size (*n*) are as indicated in each figure and legend.

## Supporting information

**S1 Fig. Computational framework. (A)** Experimental *Drosophila* imaginal disc showing cell boundaries marked with Ecad::GFP. The developing wing pouch has been segmented using ImageJ. The genotype of the *Drosophila* used is *yw;;dECad::GFP* (BL# 46556) **(B)** A pouch constructed computationally using EpiTools that served as a basis for $Ca^{2+}$ signaling simulations. In brief, cells were segmented from a wing disc. Centroids of segmented cells were used to define cellular positions in the simulated wing disc. A Voronoi tessellation followed by multiple rounds of Lloyd's relaxation [72] was used to define a template wing disc that matches the experimentally observed network topology.
(TIF)

**S2 Fig. Single-cell $Ca^{2+}$ dynamics.** The model was calibrated to match experimental single-cell frequency and amplitude. Perturbations to stimulation strength $V_{PLC}$ **(A)** alters the frequency and amplitude of $Ca^{2+}$ oscillations whereas perturbations to $k_\tau$ **(B)** only alters the frequency.
(TIF)

**S3 Fig. Bifurcation analysis of the modified model. (A)** Bifurcation diagram for the modified model used in this study; shown are the maxima and minima of the $Ca^{2+}$ oscillations (dots) and the $Ca^{2+}$ steady states (solid and dashed lines) as a function of the stimulus ($V_{PLC}$). Solid and dashed lines in red indicate stable and unstable states, respectively. Red dots indicate the maxima and minima of unstable limit cycle and the black dots indicate maxima and the minima of the stable limit cycle. HB, Hopf bifurcation occurs when $V_{PLC}$ is varied. Inset figure shows the period of $Ca^{2+}$ oscillations as a function of $V_{PLC}$. **(B)** Blocking permeability of $IP_3$, $F_P$ via gap junctions decreases $V_{PLC}$ where the initial Hopf bifurcation point ($HB_1$) occurs in

the bifurcation diagram. Block dots indicate conditions where permeability of $Ca^{2+}$, $F_c$ is set to 0. Red diamonds indicate conditions where $F_c$ is set to 0.5.
(TIF)

**S4 Fig. Model sensitivity analysis to parameters: $k_\tau$, $\tau_{max}$, $F_p$, $F_c$, $K_r$ and $K_p$.** Five different parameters in the 2D model were varied from their baseline values (BV). $V_{PLC}$ profiles of simulated tissues were selected to generate intercellular waves (red box) and are identical across all simulations to enable comparisons. Each row represents one parameter being varied in a scaling manner by fixed percentages listed in each column (*i.e.*, 50% of a BV of 1.5 μM would simulate a value of 0.75 μM). Simulations were performed varying only one parameter while holding all others constant at their BVs. Signal frequency is observed through number of bands in the kymograph, and signal duration is observed through thickness of the bands in the kymograph. **(A)** Decreased $k_\tau$ (BV of 1.5 μM) increased frequency and decreased duration of the $Ca^{2+}$ signal whereas increased $k_\tau$ did not influence the signal. **(B)** Decreased $\tau_{max}$ (BV of 800 s⁻¹) increased frequency and decreased duration of the $Ca^{2+}$ signal whereas increased $\tau_{max}$ decreased frequency and increased duration. **(C)** Decreased gap junction (GJ) communication $F_{p/c}$ (BVs of 0.005 μM² s⁻¹ for $F_p$; 0.0005 μM² s⁻¹ for $F_c$) decreased propagation of the $Ca^{2+}$ signal whereas increased GJ communication increased propagation. Signal propagation is visualized by the uniformity of the signal across the tissue. **(D)** Decreased $K_r$ (BV of 0.4 μM) decreased frequency and duration of the $Ca^{2+}$ signal whereas increased $K_r$ decreased frequency but increased duration. **(E)** Decreased $K_p$ (BV of 0.13 μM) increased frequency of the $Ca^{2+}$ signal whereas increased $K_p$ decreased frequency.
(TIF)

**S5 Fig. Model sensitivity analysis to parameters: $k_{5,P}$, $K_a$, $K_{PLC}$, $V_{SERCA}$ and $β$.** Five different parameters in the 2D model were varied from their baseline values (BV). $V_{PLC}$ profiles of simulated tissues were selected to generate intercellular waves (red box) and are identical across all simulations to enable comparisons. Each row represents one parameter being varied in a scaling manner by fixed percentages listed in each column (*i.e.*, 50% of a BV of 0.66 s⁻¹ would simulate a value of 0.33 s⁻¹). Simulations were performed varying only one parameter while holding all others constant at their BVs. Signal frequency is observed through number of bands in the kymograph, and signal duration is observed through thickness of the bands in the kymograph. **(A)** Decreased $k_{5,P}$ (BV of 0.66 s⁻¹) increased frequency of the $Ca^{2+}$ signal whereas increased $k_{5,P}$ decreased frequency. **(B)** Decreased $K_a$ (BV of 0.08 μM) increased frequency of the $Ca^{2+}$ signal to the point of observing constant activity, whereas increased $K_a$ decreased frequency to the point of loss of signal in a 60 minute simulation. **(C)** Decreased $K_{PLC}$ (BV of 0.2 μM) increased frequency of the $Ca^{2+}$ signal whereas increased $K_{PLC}$ decreased frequency to the point of loss of signal in a 60 minute simulation. **(D)** Decreased $V_{SERCA}$ (BV of 0.9 μM s⁻¹) increased frequency and duration of the $Ca^{2+}$ signal to the point of observing constant activity, whereas increased $V_{SERCA}$ decreased frequency to the point of loss of signal in a 60 minute simulation. **(E)** Decreased $β$ (BV of 0.185) increased frequency and duration of the $Ca^{2+}$ signal to the point of observing constant activity, whereas increased $β$ decreased frequency to the point of loss of signal in a 60 minute simulation.
(TIF)

**S6 Fig. Model sensitivity analysis to parameters: $K_{SERCA}$, $c_{tot}$, $k_1$ and $k_2$.** Four different parameters in the 2D model were varied from their baseline values (BV). $V_{PLC}$ profiles of simulated tissues were selected to generate intercellular waves (red box) and are identical across all simulations to enable comparisons. Each row represents one parameter being varied in a scaling manner by fixed percentages listed in each column (*i.e.*, 50% of a BV of 0.1 μM would

simulate a value of 0.05 μM). Simulations were performed varying only one parameter while holding all others constant at their BVs. Signal frequency is observed through number of bands in the kymograph, and signal duration is observed through thickness of the bands in the kymograph. **(A)** Decreased $K_{SERCA}$ (BV of 0.1 μM) decreased frequency of the $Ca^{2+}$ signal whereas increased $K_{SERCA}$ increased frequency. **(B)** Decreased $c_{tot}$ (BV of 2 μM) decreased frequency of the $Ca^{2+}$ signal whereas increased $c_{tot}$ increased frequency. **(C)** Decreased $k_1$ (BV of 1.11 s$^{-1}$) decreased frequency and duration of the $Ca^{2+}$ signal whereas increased $k_1$ increased frequency and increased. **(D)** Decreased $k_2$ (BV of 0.0203 s$^{-1}$) decreased frequency of the $Ca^{2+}$ signal whereas increased $k_2$ increased frequency.
(TIF)

**S7 Fig. Model sensitivity analysis to extremes of the parameters: $F_p$, $F_c$ and $K_{PLC}$.** Two different parameters in the 2D model were varied from their baseline values (BV). $V_{PLC}$ profiles of simulated tissues were selected to generate intercellular waves (red box) and are identical across all simulations to enable comparisons. Each row represents one parameter being varied in a scaling manner by fixed percentages listed in each column (*i.e.*, 50% of a BV of 0.1 μM would simulate a value of 0.05 μM). Simulations were performed varying only one parameter while holding all others constant at their BVs. Signal frequency is observed through number of bands in the kymograph, and signal duration is observed through thickness of the bands in the kymograph. **(A)** GJ permeability of $IP_3$ and $Ca^{2+}$ influences synchronization of $Ca^{2+}$ signaling among cells. Decreased gap junction (GJ) communication $F_{p/c}$ (BVs of 0.005 μM$^2$ s$^{-1}$ for $F_p$; 0.0005 μM$^2$ s$^{-1}$ for $F_c$) results in a transition of intercellular waves to intercellular transients, and to single-cell spikes. Increased GJ communication increased $Ca^{2+}$ signal propagation and decreased signal frequency. Signal propagation is visualized by the uniformity of the signal across the tissue. **(B)** Variations in the half-activation of $V_{PLC}$ term, $K_{PLC}$ (BV of 0.2 μM), only changed the frequency of the ICWs.
(TIF)

**S8 Fig. Schematic of expression pattern using the Gal4/UAS system. (A)** The GAL4/UAS system was used to express *GCaMP6f* transgene along with other transgenes. **(B)** *nubbin* is expressed in the wing disc pouch and the adult wing phenotype provide a readout of final phenotype after transgene expression in the wing disc pouch during larval stage.
(TIF)

**S9 Fig. Overexpression of Gαq decreases cell number and cell size. (A-B)** Wings from adult males expressing $RyR^{RNAi}$ and wild type Gαq splice 3 variant with *nubbin-Gal4, UAS GCaMP6f, UAS mcherry*. **(A'-B')** Region of interest (ROI) where the total number of setae was calculated. **(C-F)** Quantification of the wing size defined here as the area bounded by LIII, LIV, ACV and the wing margin, total cell number and cell area. Overexpression of Gαq in the pouch results in a decrease in total wing area, cell number and cell size. 10 samples were analyzed per condition. Error bars represent standard deviation. **(G)** Quantification of roundness of the adult wing. Gαq overexpression does not affect the roundness. Student t-test was used for statistical significance testing.
(TIF)

**S10 Fig. GJ inhibition is the key driver of single-cell $Ca^{2+}$ spike activity.** To replicate the ex vivo observations of insulin inducing single-cell $Ca^{2+}$ spikes, GJ permeability and the half-activation of $V_{PLC}$ were varied simultaneously in silico. **(A)** A baseline intercellular wave was used as the comparison for how parameter variations changed signal with the following parameter values: $IP_3$ gap junction permeability ($F_p$) of 0.005 μM$^2$s$^{-1}$, $Ca^{2+}$ gap junction permeability ($F_c$) of 0.0005 μM$^2$s$^{-1}$, and $K_{PLC}$ of 0.20 μM. **(B-E)** $K_{PLC}$ is increased left-to-right (red bar), and gap junction communication is

decreased left-to-right (blue bar). An increase in $K_{PLC}$ results in a decrease in frequency, while decrease in gap junction communication results in single-cell $Ca^{2+}$ spikes (red arrows).
(TIF)

**S11 Fig. Sponging cytosolic $Ca^{2+}$ increases overall wing size. (A-F)** Wings from adult males with the indicated crosses. **(A)** *nubbin-GAL4* x *UAS-RyR$^{RNAi}$* (i.e., nub4>RyR$^{RNAi}$), **(B)** *nubbin-GAL4, UAS-GCaMP6f* x *UAS-RyR$^{RNAi}$* (i.e., nub4>GCaMP6f, RyR$^{RNAi}$), **(C)** *nubbin-GAL4* x *UAS-Gaq$^{OE}$* embryonic splice 3 variant of Gaq (i.e., nub4>Gaq$^{OE}$), **(D)** *nubbin-GAL4, UAS-GCaMP6f x UAS-Gaq$^{OE}$* (i.e., nub4>GCaMP6f, Gaq$^{OE}$), **(E)** *nubbin-GAL4 x UAS-InsR$^{CA}$* (i.e., nub4>InsR$^{CA}$) gain of function mutant where the α subunit is partially deleted. **(F)** *nubbin-GAL4, UAS-GCaMP6f x InsR$^{CA}$* (i.e., nub4>GCaMP6f, InsR$^{CA}$). **(G)** Quantification of adult wings. The genetic encoded calcium sensor, GCaMP6f, binds to $Ca^{2+}$ with high affinity, thus expression of the sensor will to some degree act as a sponge of cytosolic $Ca^{2+}$. Interestingly, the presence of the GCaMP6f sponge with constitutively activated insulin signaling increases the adult wing size (E, F). Similar enhancement of wing size was observed in control wings when GCaMP6f sensor was expressed (A, B). No significant change in the adult wing size was observed when Gαq was overexpressed, suggesting sponging effects are trivialized under Gαq overexpression (C,D). Unpaired student t-test was used, and the p-values are indicated above.
(TIF)

**S12 Fig. Increasing gene dosage of GCaMP6f $Ca^{2+}$ sensor dramatically reduces wing size. (A-E)** Wings from adult males of indicated genotypes **(A)** nubG4, UAS RyR$^{RNAi}$, **(B)** nubG4, UAS GCaMP6f, **(C)** nubG4, UAS GCaMP6f/UAS GCaMP6f, **(D)** nubG4, UAS GCaMP6f (Homozygous) **(E)** nubG4, UAS CaMKII **(F)** Quantification of adult wing sizes. As the gene dose of GCaMPf is increased in the wing disc, the overall wing area decreases in size (B, C, D). Overexpressing possible $Ca^{2+}$ downstream target CaMKII increases the wing size consistent with B in which one copy of GCaMP6f was expressed.
(TIF)

**S13 Fig. Repeated simulations result in the same conclusions.** Simulations corresponding to the main text figure conclusions (*i.e*., Figs 2–4) were repeated five separate times with five different random number generator seeds. The random number generator seed value determines which cells in a simulated tissue are determined to be initiator cells. For the case of Fig 2, the $V_{PLC}$ values of standby cells also rely on the random number generator seed as the values are sampled randomly from a uniform distribution with set boundaries. The simulations' resulting video and image outputs were randomized and their inputs were hidden to allow classification of the $Ca^{2+}$ activity. **(A)** Using a graphical user interface (GUI) in MATLAB, the randomized video and kymograph simulations were drawn to show the output kymograph and play the video simulation. A user was tasked to classify the activity of the simulation output as having no activity, single-cell spikes, intercellular transient activity (ICT), intercellular wave activity (ICW), or global fluttering activity. For each main text figure, the five separate simulations had their $Ca^{2+}$ activity classified in four independent runs. Each run corresponds to a brand new running of the classification GUI, each with a different randomization scheme to display the outputs of the simulations. **(B)** The proportions of $Ca^{2+}$ activity are plotted for Fig 2's repeated simulations. Runs 1, 2, and 4 all had the same proportions, indicating reproducibility of the simulations' outputs. Run 3 had mismatched classifications between the ICT and spike class (red arrow), however, the difference in proportions was not significant using a proportions test without a continuity correction [73–75]. **(C)** The proportions of $Ca^{2+}$ activity are plotted for Fig 3's repeated simulations. Because Fig 3 was designed to demonstrate either no activity

in the case of enabled gap junction communication, or spiking activity in the case of disabled gap junction communication, only two classes of activity appear. In each classification run, there were no differences in the user-recorded classifications. **(D)** The proportions of $Ca^{2+}$ activity are plotted for Fig 4's repeated simulations. Runs 1, 2, and 4 all had the same proportions, indicating reproducibility of the simulations' outputs. Run 3 had mismatched classifications between the ICT and spike class (red arrow), however, the difference in proportions was not significant using a proportions test without a continuity correction.
(TIF)

**S1 Table. Extended data movies.**
(DOCX)

**S2 Table. Changes in wing area for known perturbations through Gal4/UAS system.** Note that maximal deviations wing size for strong growth perturbations is in range of 20–50%. References for this table include this study, [19], [76], [77].
(DOCX)

**S1 Movie. nub-Gal4>UAS-GCaMP6f, UAS-mcherry, ex vivo, spike.**
(AVI)

**S2 Movie. nub-Gal4>UAS-GCaMP6f, UAS-mcherry, ex vivo, ICT.**
(AVI)

**S3 Movie. nub-Gal4>UAS-GCaMP6f, ex vivo, ICW.**
(MP4)

**S4 Movie. nub-Gal4>UAS-GCaMP6f, ex vivo, fluttering.**
(MP4)

**S5 Movie. nub-Gal4>UAS-GCaMP6f, in vivo, spikes.**
(AVI)

**S6 Movie. nub-Gal4>UAS-GCaMP6f, in vivo, ICT.**
(MP4)

**S7 Movie. nub-Gal4>UAS-GCaMP6f, in vivo, ICW.**
(AVI)

**S8 Movie. nub-Gal4>UAS-GCaMP6f, in vivo, fluttering.**
(AVI)

**S9 Movie. Spike, Simulation output.**
(MP4)

**S10 Movie. ICT, Simulation output.**
(MP4)

**S11 Movie. ICW, Simulation output.**
(MP4)

**S12 Movie. Fluttering, Simulation output.**
(MP4)

**S13 Movie. nub-Gal4>UAS-GCaMP6f, ex vivo in Grace's low 20E media, gap junctions not blocked (Control).**
(AVI)

**S14 Movie. nub-Gal4>UAS-GCaMP6f, ex vivo in Grace's low 20E media with Carbenoxolone, gap junctions blocked.**
(AVI)

**S15 Movie. nub-Gal4>UAS-GCaMP6f, UAS-RyRRNAi, ex vivo in Grace's low 20E media (Control).**
(AVI)

**S16 Movie. nub-Gal4>UAS-GCaMP6f, UAS-InsRCA, ex vivo in Grace's low 20E media.**
(AVI)

**S17 Movie. nub-Gal4>UAS-GCaMP6f, UAS-InsRDN, ex vivo in Grace's low 20E media.**
(AVI)

**S18 Movie. nub-Gal4>UAS-GCaMP6f, UAS-GaqOE, ex vivo in Grace's low 20E media.**
(AVI)

## Acknowledgments

The authors gratefully acknowledge the Notre Dame Center for Research Computing (CRC) for providing computational facilities. The authors would also like to thank members of the Zartman lab for helpful discussions.

## Author Contributions

**Conceptualization:** Dharsan K. Soundarrajan, Francisco J. Huizar, Ramezan Paravitorghabeh, Jeremiah J. Zartman.

**Data curation:** Dharsan K. Soundarrajan, Francisco J. Huizar, Ramezan Paravitorghabeh, Trent Robinett, Jeremiah J. Zartman.

**Formal analysis:** Dharsan K. Soundarrajan, Francisco J. Huizar, Ramezan Paravitorghabeh, Jeremiah J. Zartman.

**Funding acquisition:** Jeremiah J. Zartman.

**Investigation:** Dharsan K. Soundarrajan, Francisco J. Huizar, Trent Robinett, Jeremiah J. Zartman.

**Methodology:** Dharsan K. Soundarrajan, Francisco J. Huizar, Ramezan Paravitorghabeh, Trent Robinett, Jeremiah J. Zartman.

**Project administration:** Jeremiah J. Zartman.

**Resources:** Jeremiah J. Zartman.

**Software:** Dharsan K. Soundarrajan, Francisco J. Huizar, Ramezan Paravitorghabeh.

**Supervision:** Jeremiah J. Zartman.

**Validation:** Jeremiah J. Zartman.

**Visualization:** Dharsan K. Soundarrajan, Francisco J. Huizar, Ramezan Paravitorghabeh.

**Writing – original draft:** Dharsan K. Soundarrajan, Francisco J. Huizar, Ramezan Paravitorghabeh, Jeremiah J. Zartman.

**Writing – review & editing:** Dharsan K. Soundarrajan, Francisco J. Huizar, Ramezan Paravitorghabeh, Trent Robinett, Jeremiah J. Zartman.

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
