## [Decision Letter · Decision Letter 0]

8 Jun 2021

Dear Dr. Zartman,

Thank you very much for submitting your manuscript "From spikes to intercellular waves: tuning intercellular Ca2+ signaling dynamics modulates organ size control" for consideration at PLOS Computational Biology.

As with all papers reviewed by the journal, your manuscript was reviewed by members of the editorial board and by several independent reviewers. In light of the reviews (below this email), we would like to invite the resubmission of a significantly-revised version that takes into account the reviewers' comments.

We cannot make any decision about publication until we have seen the revised manuscript and your response to the reviewers' comments. Your revised manuscript is also likely to be sent to reviewers for further evaluation.

Sincerely,

David Umulis

Associate Editor

PLOS Computational Biology

Jason Haugh

Deputy Editor

PLOS Computational Biology

Reviewer's Responses to Questions

**Comments to the Authors:**

Reviewer #1: My comments are uploaded as an attachment.

Reviewer #2: The review is uploaded as an attachment

**Have the authors made all data and (if applicable) computational code underlying the findings in their manuscript fully available?**

Reviewer #1: None

Reviewer #2: **No: **The code is available on Github and easily accessible. Some more information to be provided in order to enable its running by an independent user. Also some more details in the Supplementary Information on the algorithm/construction of the cell-based model on the computer would be useful.

PLOS authors have the option to publish the peer review history of their article (what does this mean?). If published, this will include your full peer review and any attached files.

Reviewer #1: No

Reviewer #2: No
---

## [Decision Letter · Decision Letter 1]

28 Sep 2021

Dear Dr. Zartman,

Thank you very much for submitting your manuscript "From spikes to intercellular waves: tuning intercellular Ca2+ signaling dynamics modulates organ size control" for consideration at PLOS Computational Biology. As with all papers reviewed by the journal, your manuscript was reviewed by members of the editorial board and by several independent reviewers. The reviewers appreciated the attention to an important topic. Based on the reviews, we are likely to accept this manuscript for publication, providing that you modify the manuscript according to the review recommendations. If so, I anticipate a rapid decision on the R2 manuscript.

Sincerely,

Jason M. Haugh

Deputy Editor

PLOS Computational Biology

[LINK]

Reviewer's Responses to Questions

**Comments to the Authors:**

Reviewer #1: The authors have clearly addressed my questions. I have no more comments.

Reviewer #2: See attachment

**Have the authors made all data and (if applicable) computational code underlying the findings in their manuscript fully available?**

Reviewer #1: None

Reviewer #2: Yes

PLOS authors have the option to publish the peer review history of their article (what does this mean?). If published, this will include your full peer review and any attached files.

Reviewer #1: No

Reviewer #2: No

Figure Files:

Data Requirements:

Reproducibility:

References:

---

## [Editor Report · Decision Letter 2]

7 Oct 2021

Dear Dr. Zartman,

We are pleased to inform you that your manuscript 'From spikes to intercellular waves: tuning intercellular Ca2+ signaling dynamics modulates organ size control' has been provisionally accepted for publication in PLOS Computational Biology.

Best regards,

Jason M. Haugh

Deputy Editor

PLOS Computational Biology

---

## [Editor Report · Acceptance letter]

25 Oct 2021

PCOMPBIOL-D-21-00695R2 

From spikes to intercellular waves: tuning intercellular calcium signaling dynamics modulates organ size control

Dear Dr Zartman,

I am pleased to inform you that your manuscript has been formally accepted for publication in PLOS Computational Biology. Your manuscript is now with our production department and you will be notified of the publication date in due course.

With kind regards,

Olena Szabo
